# Immune Control of Human Cytomegalovirus (HCMV) Infection in HCMV-Seropositive Solid Organ Transplant Recipients: The Predictive Role of Different Immunological Assays

**DOI:** 10.3390/cells13161325

**Published:** 2024-08-08

**Authors:** Federica Zavaglio, Irene Cassaniti, Piera d’Angelo, Paola Zelini, Giuditta Comolli, Marilena Gregorini, Teresa Rampino, Lucia Del Frate, Federica Meloni, Carlo Pellegrini, Massimo Abelli, Elena Ticozzelli, Daniele Lilleri, Fausto Baldanti

**Affiliations:** 1Microbiology and Virology Unit, Fondazione IRCCS Policlinico San Matteo, 27100 Pavia, Italy; f.zavaglio@smatteo.pv.it (F.Z.); p.dangelo@smatteo.pv.it (P.d.); p.zelini@smatteo.pv.it (P.Z.); g.comolli@smatteo.pv.it (G.C.); d.lilleri@smatteo.pv.it (D.L.); f.baldanti@smatteo.pv.it (F.B.); 2Department of Clinical, Surgical, Diagnostic and Pediatric Sciences, University of Pavia, 27100 Pavia, Italy; c.pellegrini@smatteo.pv.it; 3Unit of Nephrology, Dialysis and Transplantation, Fondazione IRCCS Policlinico San Matteo, 27100 Pavia, Italy; m.gregorini@smatteo.pv.it (M.G.); t.rampino@smatteo.pv.it (T.R.); 4Department of Internal Medicine and Therapeutics, University of Pavia, 27100 Pavia, Italy; 5Transplant Centre Unit, Fondazione IRCCS Policlinico San Matteo, 27100 Pavia, Italy; l.delfrate@smatteo.pv.it (L.D.F.); f.meloni@smatteo.pv.it (F.M.); 6Cardiac Surgery, Department of Intensive Medicine, Fondazione IRCCS Policlinico San Matteo, 27100 Pavia, Italy; 7Department of Surgery, University of Pavia, Fondazione IRCCS Policlinico San Matteo, 27100 Pavia, Italy; m.abelli@smatteo.pv.it (M.A.); e.ticozzelli@smatteo.pv.it (E.T.)

**Keywords:** human cytomegalovirus, T cell response, solid organ transplant recipients, immune assays

## Abstract

Human cytomegalovirus (HCMV) infection remains a major complication for solid organ transplant recipients (SOTRs). The aim of this study was to evaluate the role of HCMV-specific T cell immunity measured at the time of the HCMV-DNA peak in predicting the spontaneous clearance of infection. The performance of cytokine flow cytometry using infected dendritic cells (CFC-iDC), infected cell lysate (CFC-iCL) and pp65 peptide pool (CFC-pp65 pool) as stimuli, as well as ELISPOT assays using infected cell lysate (ELISPOT-iCL) and the pp65 peptide pool (ELISPOT-pp65 pool), was analysed. Among the 40 SOTRs enrolled, 16 patients (40%) required antiviral treatment for an HCMV infection (Non-Controllers), while the others spontaneously cleared the infection (Controllers). At the HCMV-DNA peak, the number of HCMV-specific CD4^+^ T cells detected by the CFC-iDC, CFC-iCL and CFC-pp65 pool assays in Controllers was higher than that detected in Non-Controllers, while no difference was observed in terms of HCMV-specific CD8^+^ T cell response. The same trend was observed when the HCMV-specific T cell response was measured by ELISPOT-iCL and ELISPOT-pp65 pool. We observed that the CD4^+^ CFC-pp65 pool assay was the best predictor of self-resolving HCMV infection at the time of the HCVM-DNA peak. The CFC-pp65 pool assay is able to discriminate between CD4^+^ and CD8^+^ T cell responses and could be used in daily clinical practice.

## 1. Introduction

*Human cytomegalovirus* (HCMV) infection remains a concerning opportunistic infection in solid organ transplant recipients (SOTRs) and in patients with hematologic malignancies or other immunocompromising conditions, as it is associated with significant morbidity and mortality [1,2,3]. Currently, two strategies have been adopted for preventing HCMV disease: a universal prophylaxis (i.e., the administration of antiviral drugs to all transplanted patients for 6–12 months) and active surveillance with pre-emptive therapy administration (i.e., monitoring the blood viral load and giving antiviral drugs to patients at predetermined levels of viral load) [4,5]. Moreover, it is conceivable that patients with a sustained HCMV-specific T cell response may benefit from reduced antiviral therapy and surveillance [6,7,8,9]. In recent years, it has been shown that the quantification of cellular in vitro responses against HCMV can help to stratify the risk of HCMV disease [10,11,12,13,14,15,16,17,18,19,20,21,22,23]. In a previous study, three different immunological assays for the quantification of HCMV-specific T cell responses were evaluated in order to determine their potential role in predicting protection against HCMV disease [14]. However, the timing of the immunological monitoring and the actual predictive role of these assays in standard diagnostic procedures for the management of HCMV infections in SOTRs need to be further discussed.

Thus, the aim of our study was to evaluate the role of HCMV-specific T cell immunity measured at the time of the HCMV-DNA peak in predicting the spontaneous clearance of HCMV infection. The performance of cytokine flow cytometry (CFC) using infected dendritic cells (CFC-iDC), infected cell lysate (CFC-iCL) and pp65 peptide pool (CFC-pp65 pool) as stimuli, as well as ELISPOT assays using infected cell lysate (ELISPOT-iCL) and pp65 peptide pool (ELISPOT-pp65 pool), was analysed.

## 2. Material and Methods

### 2.1. Study Subjects

Between June 2018 and December 2019, eighty-six SOTRs were consecutively transplanted at Fondazione IRCCS Policlinico San Matteo, Pavia, Italy. A total of 75 of them were HCMV-seropositive and 11 were HCMV-seronegative; only 4 SOTRs did not develop HCMV DNAemia. Among the 86 SOTRs, 40 HCMV-seropositive SOTRs, including 27 kidney transplant recipients (KTRs), 6 lung transplant recipients (LTRs) and 7 heart transplant recipients (HTRs), had available HCMV-specific T-cell immunity tests and were analysed in this study. Their clinical and demographic characteristics are included in Table 1. HCMV serological status was positive in 22 donors, negative in 13 donors and unknown in 5 donors. According to local hospital protocols, HCMV DNAemia was quantified twice weekly by real-time PCR during the first three months after transplant. Later, patients were monitored for HCMV DNAemia at scheduled medical visits (at least once/month) or in the presence of HCMV-related clinical symptoms.

Patients were defined as “Controllers” in cases of a self-resolving HCMV infection and no sign of HCMV disease or as “Non-Controllers” if they required treatment for a systemic HCMV infection and/or diagnosed tissue-invasive disease (TID). Antiviral treatment with ganciclovir (GCV) or valganciclovir (VGCV) was administered pre-emptively, after the detection of 300,000 HCMV-DNA copies/mL whole blood (corresponding to 120,000 international units [IU]/mL [24]), or in cases of suspected or diagnosed TID. In detail, 3 Non-Controllers had TID and no patients had HCMV syndrome. Only one patient experienced graft rejection. The HCMV-specific T cell response was evaluated at the HCMV-DNA peak (between 2 and 3 months after transplant) to predict the spontaneous clearance of HCMV infection.

The study was approved by the Ethics Committee and Fondazione IRCCS Policlinico San Matteo Institutional Review Board (Procedure No. 201800034325) and the patients gave written informed consent.

### 2.2. CFC-iDC, CFC-iCL and CFC-pp65 Pool Assays and Cytokine Flow Cytometry Staining

HCMV-specific CD4^+^ and CD8^+^ T cells from peripheral blood mononuclear cells (PBMCs) were stimulated using autologous, monocyte-derived, HCMV VR1814-infected dendritic cells (iDCs), as previously reported [25]; then, both activated CD4^+^ and CD8^+^ T cells were quantified by a cytokine flow cytometry analysis of intracellular interferon-γ (IFN-γ) production [26]. In parallel, a commercially available HCMV AD169-infected cell lysate (iCL; Microbix Biosystem, Missisauga, ON, Canada) at a final concentration of 50 µg/mL [14] and a pp65 peptide pool (JPT, Peptide Technologies, Berlin, Germany) at a final concentration of 1 µg/mL were used as stimuli for quantifying activated HCMV-specific CD4^+^ and CD8^+^ T cells. In CFC-iDC assays, non-infected dendritic cells were used as a negative control, while, for the CFC-iCL and CFC-pp65 pool assays, human actin (15 mers, overlapping by 10 amino acids, Pepscan, Lelystad, The Netherlands), at a final concentration of 1 µg/mL, was used as a negative control. Following 24 h incubation with iDCs, iCL or pp65 peptide pool and the appropriate control, PBMCs were stained for intracellular IFN-γ production with the CFC assay [25,26]. Cells were washed with PBS 2 mM EDTA and stained in PBS with Live/Dead Fixable Pacific Blue (Invitrogen, Waltham, MA, USA) for 30 min at 4 °C. After rising with PBS, cells were stained in PBS 5% FBS with CD8 V500 (BD Biosciences, Franklin Lakes, NJ, United States) for 30 min at 4 °C. Cells were then washed with PBS 5% FBS, fixed and permeabilised using Citofix/Citoperm (BD Biosciences) for 20 min at 4 °C. Final staining was carried out with CD3 PerCP-Cy 5.5, CD4 APC-Cy7 and IFN-γ PE-Cy7 (all from BD Biosciences) antibodies in Perm/Wash (BD Biosciences) for 45 min at room temperature. Cells were then washed and resuspended in PBS 1% paraformaldehyde (Merck, Rahway, NJ, United States).

Analysis was performed with FACS Canto II and a FACS Lyric flow cytometer using BD DIVA v6.1.3 and BD FACSuite v1.5 software (all from BD Biosciences). A representative pseudocolour plot analysis is shown in Figure 1.

The frequency of IFN-γ-producing CD4^+^ and CD8^+^ T cells was determined by subtracting the frequency of T cells stimulated with the control antigen from the frequency of T cells incubated with the HCMV antigen. The absolute CD3^+^CD4^+^ and CD3^+^CD8^+^ T cell count was measured in whole blood using flow cytometry (BD Multitest^TM^ CD3/CD8/CD45/CD4 with BD TruCOUNT^TM^ Tubes, BD Biosciences). The total number of HCMV-specific CD4^+^ and CD8^+^ T cells was then calculated by multiplying the percentage of IFN-γ-producing HCMV-specific T cells by the corresponding absolute CD4^+^ and CD8^+^ T cell count.

### 2.3. ELISPOT-iCL

A 24 h ELISPOT assay was performed with a commercial CE-marked kit for IFN-γ detection (Autoimmune Diagnostika, Strassberg, Germany), modified using iCL (Microbix Biosystem) at a final concentration of 50 µg/mL as the stimulus and RPMI medium only as the negative control [27]. The results were acquired and analysed using an automated AID ELISPOT reader system (Autoimmun Diagnostika GmbH, Strassberg, Germany). The net spots per million of PBMCs were calculated by subtracting the number of spots responding to the negative control from the number of spots responding to the corresponding antigen and the results were given as net spots/million PBMCs. The total number of HCMV-specific spot-forming T cells/µL was calculated with the formula (net spot number × lymphocyte number/µL blood)/2 × 10^5^ [27]. A representative image of an ELISPOT assay is shown in Figure 2.

### 2.4. ELISPOT-pp65 Peptide Pool

Human IFN-*γ* ELISPOT kits (Diaclone, Besancon, France) and Multiscreen-IP membrane-bottomed 96-well plates (Merck Millipore, Darmstadt, Germany) were used for a 24 h ELISPOT assay modified by stimulating the cells with a commercially available pp65 peptide pool (JPT, Peptide Technologies, Berlin, Germany) at a final concentration of 0.25 µg/mL and RPMI medium only (negative control), as described previously [14,27]. Spots were counted by using an automated AID ELISPOT reader system (Autoimmun Diagnostika GmbH). The net spots per million PBMCs were calculated by subtracting the number of spots responding to the negative control from the number of spots responding to the corresponding antigen and the results were given as net spots/million PBMCs. The total number of HCMV-specific spot-forming T cells/µL was calculated with the formula (net spot number × lymphocyte number/µL blood)/2 × 10^5^. A representative image of an ELISPOT assay is shown in Figure 2.

### 2.5. Statistical Analysis

Quantitative variables are shown as medians and interquartile ranges (IQRs) while qualitative variables are shown as frequencies or absolute numbers. The Mann–Whitney U-test was applied for unpaired comparisons. Receiver–operator characteristic (ROC) analysis was used to assess the performance of the five immunological assays in predicting the spontaneous control of an HCMV infection. The area under the curve (AUC) and its 95% confidence interval (CI) were calculated. The best cut-off indicating protection from HCMV infection for the five assays was calculated according to the Youden index. All the analyses were performed using GraphPad Prism 8.3.0 (GraphPad Software Inc., La Jolla, CA, USA). All the tests were two-tailed and a *p* value < 0.05 was considered statistically significant.

## 3. Results

### 3.1. Clinical Characteristics of the Patients Enrolled

Overall, 24/40 (60%) patients were defined as Controllers while 16/40 (40%) were Non-Controllers. HCMV infection was detected after a median time of 48 days (IQR 35–73 days) in Controllers and 36 days (IQR 22–43 days) in Non-Controllers. The HCMV-DNA peak was detected after a median time of 64 days (IQR 43–95 days) in Controllers and 61 days (IQR 45–81 days) in Non-Controllers. The median level of the HCMV-DNA peak was 7470 copies/mL (IQR 1553–16,223 copies/mL; 2988 IU/mL, IQR 621.2–6489.2 IU/mL) in Controllers and 175,959 copies/mL (IQR 52,200–459,900 copies/mL; 70,383.6 IU/mL IQR 20,800–183,960 IU/mL) in Non-Controllers (*p* < 0.001; Figure 3). No differences in terms of clinical and demographic characteristics were observed in the two groups of patients, as reported in Table 1.

### 3.2. Evaluation of HCMV-Specific T Cell Response in Controllers and Non-Controllers Using Five Different Immunological Assays

At the HCMV-DNA peak (median time 70 [IQR 51–89] days after transplant), the absolute number of total CD4 and CD8 T cells was evaluated in Controllers and Non-Controllers (Figure 4A,B). No difference was observed between the two groups (Figure 4A,B). The percentage of HCMV-specific CD4^+^ T cells detected by the CFC-iDC assay in Controllers was higher than that detected in Non-Controllers (*p* = 0.011; Figure 5A); the same result was observed in terms of the HCMV-specific CD8^+^ T cell percentage (*p* = 0.043; Figure 5B). The same trend was observed when the HCMV-specific T cell response was measured by CFC-iCL assay and CFC-pp65 pool. In detail, Controllers had significantly higher levels of HCMV-specific CD4^+^ T cells measured by the CFC-iCL assay than Non-Controllers (*p* < 0.001; Figure 5C). On the other hand, looking at HCMV-specific CD8^+^ T cells, no difference was observed in the two groups (*p* = 0.270; Figure 5D). The number of CD4^+^ T cells detected by the CFC-pp65 pool assay was significantly higher in Controllers (*p* < 0.001; Figure 5E), and a trend for higher levels of HCMV-specific CD8^+^ T cells in Controllers was also observed (*p* = 0.090; Figure 5F). The HCMV-specific T cells detected by ELISPOT-iCL and pp65 pool showed a trend towards higher levels in Controllers than Non-Controllers (*p =* 0.004 and *p* = 0.001, respectively; Figure 5G,H). We also evaluated the number of HCMV-specific CD4^+^ and CD8^+^ T cells/μL blood using CFC-iDC, iCL and pp65 pool (Figure 6A–F) and the number of HCMV-specific T cells/μL blood using ELISPOT-iCL and pp65 pool (Figure 6G,H). The results were similar (Figure 6A–H) to those obtained with the percentage of HCMV-specific CD4^+^ and CD8^+^ T cells (Figure 5A–H).

### 3.3. Comparison of Prognostic Performance of Five Different Immunological Assays and Determination of Cut-off Values for Protective T Cell Response

To compare the prognostic performances of the five immunological assays (CFC-iDC, iCL and pp65 pool, as well as ELISPOT-iCL and pp65 pool) in the identification of Controllers and Non-Controllers, an ROC curve analysis was performed (Table 2 and Table 3).

When we considered the percentage of T cells, the CD4^+^ CFC-iCL and pp65 pool were the best predictors of self-resolving HCMV infection and showed the best AUC (0.81; 95% CI 0.66 to 0.96; 0.80 95% CI: 0.67 to 0.94, respectively), while the AUCs for the CD4^+^ CFC-iDC, ELISPOT-iCL and ELISPOT-pp65 pool were slightly inferior (0.73 to 0.78). The AUCs for CD8^+^ T cells measured by the CFC-iDC, CFC-iCL and CFC-pp65 pool were the worst (<0.70). The cut-offs for the CD4^+^ and CD8^+^ CFC-iDC, iCL and pp65 pool and ELISPOT-iCL and pp65 pool were calculated with the Youden index. We selected the best cut-offs considering the maximal level of sensitivity and specificity in the identification of Controllers (Table 2).

In contrast, when we considered the absolute number, the CD4^+^ CFC-pp65 pool assay was the best predictor of self-resolving HCMV infections and showed the best AUC (0.81; 95% CI: 0.66 to 0.95), while the AUCs for the CD4^+^ CFC-iDC, CFC-iCL, ELISPOT-iCL and ELISPOT-pp65 pool were slightly inferior (0.72 to 0.76). The AUCs for the CD8^+^ T cells measured by CFC-iDC, CFC-iCL or CFC-pp65 pool were the worst (<0.70).

For the CFC-iDC, CFC-iCL and CFC-pp65 pool assays, the cut-offs of 0.4 HCMV-specific CD4^+^ T cells/μL and 2 HCMV-specific CD8^+^ T cells/μL were selected as the ones showing the maximal level of sensitivity and specificity in the identification of Controllers (Table 3), while, for the ELISPOT-iCL and pp65 pool assays, the better cut-offs were 0.1 and 0.5 HCMV-specific T cells/μL (Table 3). The cut-offs for the CD4^+^ and CD8^+^ CFC-iDC, as well as the ELISPOT-iCL and pp65 pool, calculated with the Youden index in this study are in accordance with the cut-offs calculated previously [7,14,28].

The values of sensitivity and specificity obtained by the CFC-pp65 pool were in contrast to those observed with CFC-iDC. In fact, the CD4^+^ CFC-iDC showed a higher sensitivity in predicting self-resolving infection (Controller patients), while the CD4^+^ CFC-pp65 pool showed the best specificity in predicting Non-Controller patients that required treatment for an HCMV infection (Table 2, Figure 6A,G).

## 4. Discussion

The objective of this study was to evaluate the prognostic performance of five assays in the quantification of the HCMV-specific T cell response for predicting the spontaneous clearance of HCMV infection in SOTRs. The HCMV CD4^+^ T cell numbers measured by any of the assays evaluated were predictive of self-resolving HCMV infection. Particularly, if we considered the percentage of CD4^+^ T cells, the CFC-iCL pp65 pool assays were the best predictors of HCMV control, while, if we considered the absolute number of CD4^+^ T cells, the CFC-pp65 pool assays were the best predictors of HCMV control. Additionally, the percentage and the absolute number of HCMV-specific CD8^+^ T cells were poorly predictive.

In a previous study, we compared three different assays (CFC-iDC, ELISPOT-iCL and ELISPOT-pp65 pool) for the determination of an HCMV-specific T cell response that can be efficiently adopted to identify KTRs with immune protection against HCMV disease despite immune-suppressive therapy [14]. The present study further extends the analysis to LTRs and HTRs and to two additional CFC assays using iCL and pp65 pool as stimuli. The protective cut-offs defined in the present study confirm those previously determined for CFC-iDC and the two ELISPOTs [7,14,28].

The use of different immunological assays to assess the risk of HCMV DNAemia and disease has been addressed in various studies, but few of them have compared their performances. In detail, some studies reported that QuantiFERON-CMV assays can predict late-onset HCMV disease after primary prophylaxis and the spontaneous clearance of HCMV DNAemia in SOTRs [23,29,30]. However, contrasting results have been reported when the performances of the QuantiFERON and ELISPOT have been analysed [10,31]. Gliga et al. compared two commercial ELISPOT assays (T-Track CMV and T-SPOT CMV) with QuantiFERON-CMV [13], showing the lower specificity of the Quantiferon assayuantiFERON assay in identifying patients able to control spontaneously HCMV. Rogers and colleagues reported a correlation between a low number of HCMV-specific CD4^+^ T cells and subsequent HCMV events, but not with HCMV-specific CD8^+^ T cells, using Viracor^®^ CMV T Cell Immunity Panel (CMV-TCIP) [12]. Thus, these results confirm the better ability of CD4^+^ than CD8^+^ T cell measurements to identify patients with protective immunity, mostly because a long-term protection from HCMV infection is achieved when the CD4^+^ T cell response is restored [11,25,32,33,34,35,36]; moreover, CD8^+^ T cells seem to be not protective in the absence of their CD4^+^ T cell counterpart [11,32,37].

As a major limitation, the ELISPOT and QuantiFERON assays do not allow the differential quantification of CD4^+^ and CD8^+^ T cells. The QuantiFERON-CMV assay targets CD8^+^ T cells only, while the ELISPOT assays target both CD4^+^ and CD8^+^ T cells without distinguishing them [30,38]. The actual advantage of the cytokine flow cytometry assay is its ability to discriminate between CD4^+^ and CD8^+^ T cell responses [12,14].

Among the assays measuring the absolute number of HCMV-specific CD4^+^ T cells, the CD4^+^ CFC-iDC assay showed the best sensitivity in detecting the HCMV-specific T cell recovery after transplantation, while the CD4^+^ CFC-pp65 pool assay had better specificity in predicting self-resolving HCMV infections. The higher sensitivity of CFC-iDC in detecting CD4^+^ T cell recovery might be related to the use of HCMV-infected DCs as an antigen, allowing a broader antigenic stimulation than pp65, which underestimates the actual HCMV-specific T cell response when used as the sole stimulus, as already reported [39]. However, the use of iDCs requires a 7-day turnaround time and skills in cell culture and live virus handling. Conversely, although less sensitive, the CFC-pp65 pool or CFC-iCL, which are able to discriminate between CD4^+^ and CD8^+^ T cell responses, could be used in daily clinical practice because their results are reliable and can be obtained in 24 h. The ELISPOT-iCL and ELISPOT-pp65 pool are valuable alternatives that do not require specialised staff to perform flow cytometry.

Our study is limited by its retrospective and observational nature and by its relatively low sample size, which does not allow for statistical inference on the diagnostic performance of the assays used for CD4^+^ T cell measurements. Nevertheless, our study confirms the better reliability of HCMV-specific CD4^+^ than CD8^+^ determination for the identification of transplanted patients at low risk for HCMV disease. Patients with low HCMV DNA levels may have a low HCMV-specific T cell response; thus, it is likely that the virus will continue to replicate or a high T cell response will be able to control further virus replication.

## 5. Conclusions

In conclusion, our study showed that if we considered the percentage of HCMV-specific CD4^+^ T cells, the CFC-iCL and pp65 pool assays were the best predictors of self-resolving HCMV infection, while, if we considered the absolute number of HCMV-specific CD4^+^ T cells, the CFC-pp65 pool was the best predictor of self-resolving HCMV infection. Both the percentage and absolute number of HCMV-specific T cells have the same predictive value. Both the CFC-iCL and pp65 pool could be used in clinical practice and their results can be obtained in 24 h. In the future, the development of a QuantiFERON-like rapid assay for the evaluation, in whole blood, of the CD4^+^ instead of the CD8^+^ T cell response to HCMV could represent a step forward for the widespread introduction of HCMV-specific immune monitoring in routine clinical practice.

## Figures and Tables

**Figure 1 cells-13-01325-f001:**
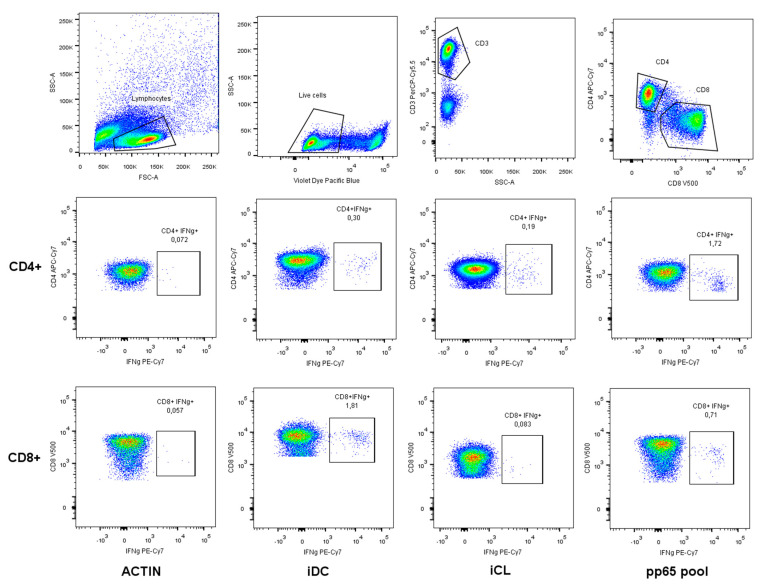
Cytokine flow cytometry (CFC) gating strategy. Representative gating of antigen-specific CD4^+^ and CD8^+^ T cells from immunocompromised patients after stimulation with human actin, infected dendritic cells (iDCs), infected cell lysate (iCL) and pp65 peptide pool. Briefly, lymphocyte cells were gated out of all events followed by live Pacific Blue cells. Cells were then gated as CD3 PerCP5.5^+^. T cells were further subdivided into CD4 APC Cy7^+^ and CD8 V500^+^ populations. T cells were defined as CD4^+^ IFN-γ^+^ or CD8^+^ IFN-γ^+^.

**Figure 2 cells-13-01325-f002:**
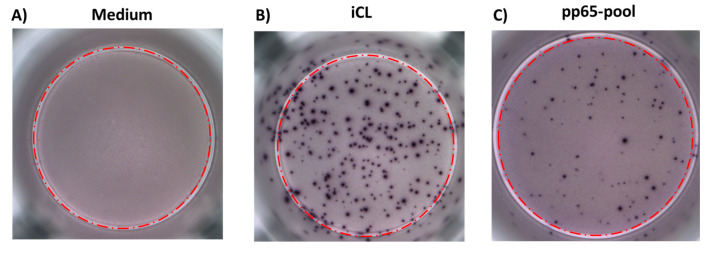
Image representation of ELISPOT assay. The black spots represent IFN-γ-producing T cells. (**A**) IFN-γ T cells in medium (as negative control); (**B**) HCMV-specific T cells after stimulation with infected cell lysate (iCL); (**C**) HCMV-specific T cells after stimulation with pp65 pool. Red dotted circles are referred to the well area.

**Figure 3 cells-13-01325-f003:**
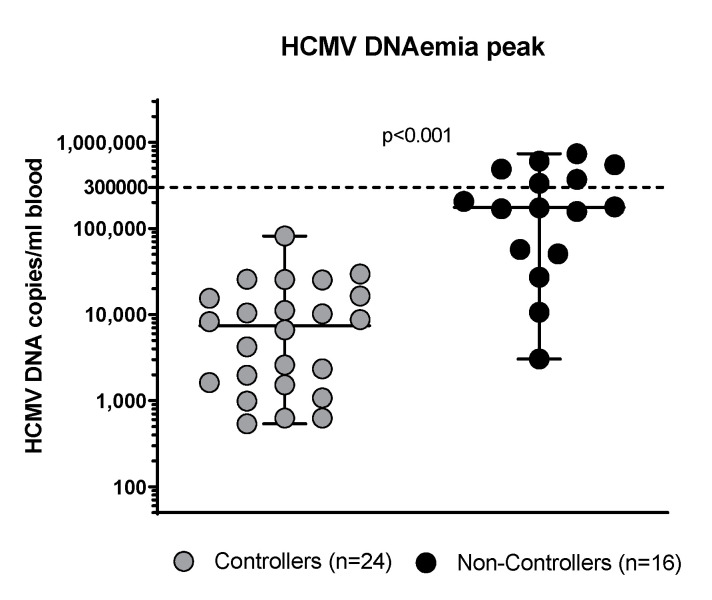
HCMV-DNA blood peak in Controllers (solid grey circles) and Non-Controllers (solid black circles).

**Figure 4 cells-13-01325-f004:**
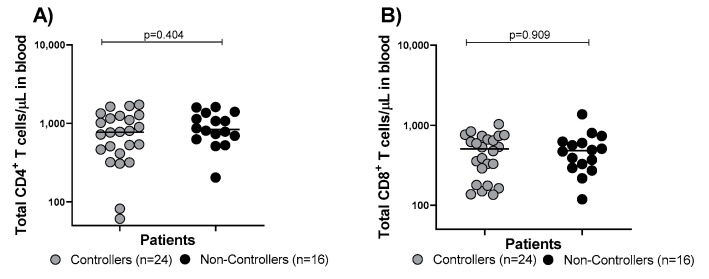
The total number of CD4^+^ (**A**) and CD8^+^ (**B**) T cells in blood was evaluated in Controllers (solid grey circles) and Non-Controllers (solid black circles) at the time of the HCMV-DNA peak.

**Figure 5 cells-13-01325-f005:**
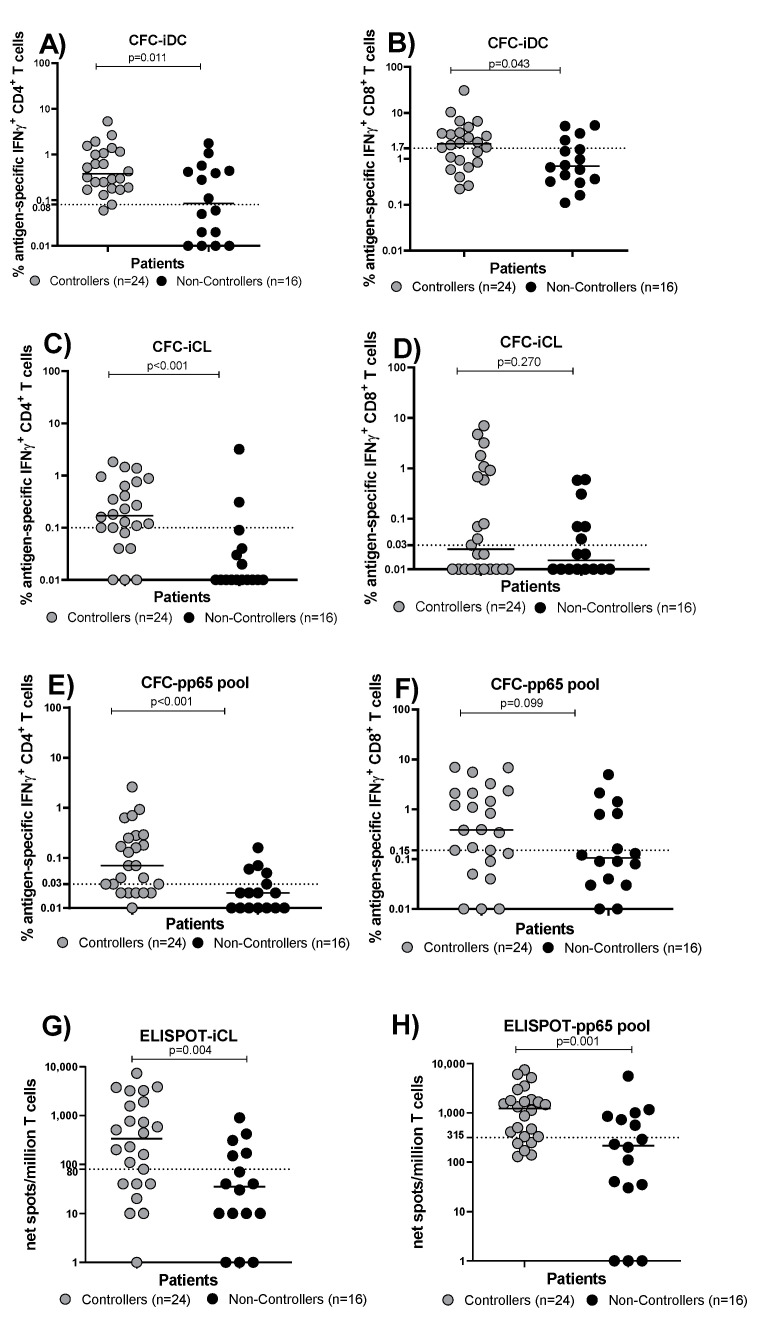
The percentage of HCMV-specific T cells was evaluated in Controllers (solid grey circles) and Non-Controllers (solid black circles) at the time of the HCMV-DNA peak using CFC-iDC, iCL and pp65 assays and ELISPOT-iCL and pp65 assays. (**A**) Percentage of IFNγ^+^ CD4^+^ T cells seen with the CFC-iDC assay. (**B**) Percentage of IFNγ^+^ CD8^+^ T cells seen with the CFC-iDC assay. (**C**) Percentage of IFNγ^+^ CD4^+^ T cell response seen with the CFC-iCL assay. (**D**) Percentage of IFNγ^+^ CD8^+^ T cells seen with the CFC-iCL assay. (**E**) Percentage of IFNγ^+^ CD4^+^ T cells seen with the CFC-pp65 pool assay. (**F**) Percentage of IFNγ^+^ CD8^+^ T cells seen with the CFC-pp65 pool assay. (**G**) Percentage of IFNγ^+^ T cells seen with the ELISPOT-iCL assays. (**H**) Percentage of IFNγ^+^ T cells seen with the ELISPOT-pp65 pool assays. The horizontal dotted line indicated the cut-off.

**Figure 6 cells-13-01325-f006:**
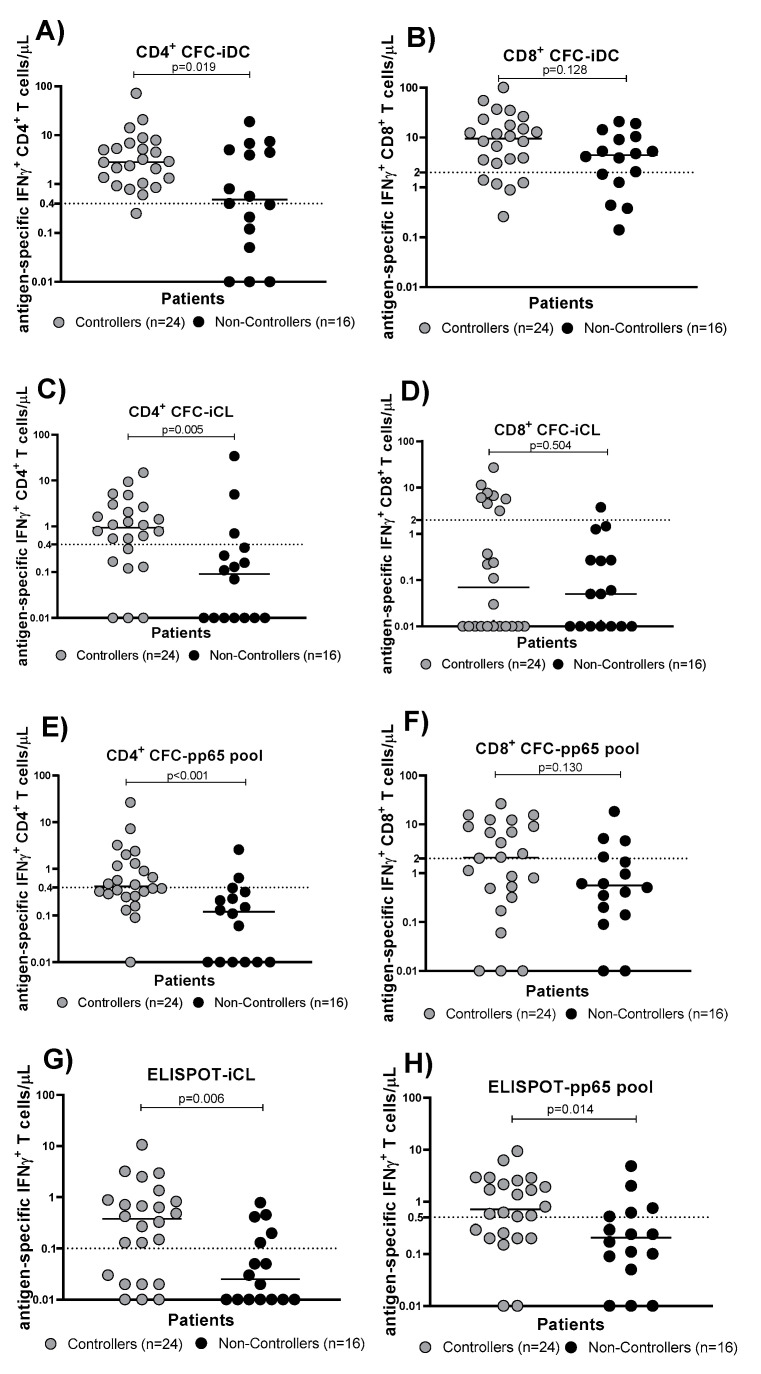
The number of HCMV-specific T cells was evaluated in Controllers (solid grey circles) and Non-Controllers (solid black circles) at the time of the HCMV-DNA peak using CFC-iDC, iCL and pp65 assays and ELISPOT-iCL and pp65 assays. (**A**) IFNγ^+^ CD4^+^ T cells/μL seen with the CFC-iDC assays. (**B**) IFNγ^+^ CD8^+^ T cells/μL seen with the CFC-iDC assays. (**C**) IFNγ^+^ CD4^+^ T cells/μL seen with the CFC-iCL assays. (**D**) IFNγ^+^ CD8^+^ T cells/μL seen with the CFC-iCL assays. (**E**) IFNγ^+^ CD4^+^ T cells/μL seen with the CFC-pp65 pool assays. (**F**) IFNγ^+^ CD8^+^ T cells/μL seen with the CFC-pp65 pool assays. (**G**) IFNγ^+^ T cells/μL seen with the ELISPOT-iCL assays. (**H**) IFNγ^+^ T cells/μL seen with the ELISPOT-pp65 pool assays. The horizontal dotted line indicated the cut-off.

**Table 1 cells-13-01325-t001:** Patients’ characteristics.

Characteristics	All Patients (*n* = 40)	Controllers (*n* = 24)	Non-Controllers (*n* = 16)	*p* Value
Age, median [IQR]	51 [45–58]	51 [43–58]	53 [45–59]	0.401
Gender, *n* (%):				
Male	28 (70)	15 (62)	13 (81)	0.290
Female	12 (30)	9 (38)	3 (19)	
Kind of Transplant, *n* (%):				
KTR	27 (67)	17 (71)	10 (63)	0.683
LTR	6 (15)	4 (17)	2 (12)	0.684
HTR	7 (18)	3 (12)	4 (25)	0.591
Induction Therapy, *n* (%):				
Basiliximab	29 (72)	17 (71)	12 (75)	0.457
ATG	7 (17)	5 (21)	2 (12)	0.418
Steroids	40 (100)	24 (100)	16 (100)	0.365
**Maintenance Therapy, *n* (%):**				
FK506, MMF, MPRE	33 (83)	20 (83)	13 (81)	0.380
CyA, MMF, MPRE	4 (10)	3 (13)	1 (7)	0.458
FK506, Everolimus, MPRE	3 (7)	1 (4)	2 (12)	0.561

**Legend**. KTR, kidney transplant recipient; LTR, lung transplant recipient; HTR, heart transplant recipient; ATG, anti-human thymocyte globulin; FK506, tacrolimus; MMF, mycophenolate mofetil; MPRE, methylprednisolone; CyA, Cyclosporine A.

**Table 2 cells-13-01325-t002:** Prognostic performances of different assays at HCMV peak considering the percentage of T cells.

	CFC-iDC CD4	CFC-iDC CD8	CFC-iCL CD4	CFC-iCL CD8	CFC-pp65 CD4	CFC-pp65 CD8	ELISPOT- iCL	ELISPOT-pp65
AUC	0.73	0.69	0.81	0.60	0.80	0.65	0.78	0.76
CUT-OFF (%)	0.08	1.70	0.10	0.03	0.03	0.15	80	315
sensitivity (%)	95	62	75	50	75	70	70	79
specificity (%)	40	75	87	62	68	62	68	64
PPV (%)	74	79	90	67	78	74	77	76
NPV (%)	89	57	70	45	65	59	61	67

AUC, area under curve; CI, confidence interval; PPV, positive predictive value; NPV, negative predictive value.

**Table 3 cells-13-01325-t003:** Prognostic performances of different assays at HCMV peak considering the absolute number of T cells.

	CFC-iDC CD4	CFC-iDC CD8	CFC-iCL CD4	CFC-iCL CD8	CFC-pp65 CD4	CFC-pp65 CD8	ELISPOT- iCL	ELISPOT-pp65
AUC	0.72	0.64	0.76	0.56	0.81	0.64	0.75	0.73
CUT-OFF	0.4	2	0.4	2	0.4	2	0.1	0.5
sensitivity (%)	95	76	71	33	50	50	71	62
specificity (%)	44	31	75	94	81	75	69	75
PPV (%)	72	66	85	89	86	76	77	76
NPV (%)	85	50	65	48	54	52	61	58

AUC, area under curve; CI, confidence interval; PPV, positive predictive value; NPV, negative predictive value.

## Data Availability

The data that support the findings of this study are available on request from the corresponding author. The data are not publicly available due to privacy or ethical restrictions.

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
