# Peer review of "Immune Control of Human Cytomegalovirus (HCMV) Infection in HCMV-Seropositive Solid Organ Transplant Recipients: The Predictive Role of Different Immunological Assays"

_cells, 2024, doi:10.3390/cells13161325_

Round 1

Reviewer 1 Report

Comments and Suggestions for Authors

This is an interesting study but with some gaps.

Some important clinical variables are missing.

All the controlors had a number of DNA copies under 300.000 and without suspected or diagnosed tissue-invasive disease?

It’s important to know the donor and receptor CMV serological status

Immunosuppressive regimen

Table 1: FK506 25 patients and tacrolimus 10 patients is the same drug ?!

Normally the KT patients are under triple immunosupression: CNI & MMF/MPA & steroid ; CNI & steroid & mTOR inhibitor for example – i don’t understand this table

Other important variables

Patients with end-organ CMV disease

Patients with CMV syndrome

Time to onset of CMV event

Graft rejection or other event with change in Immunosuppressive regimen

Author Response

Comments and Suggestions for Authors.

 This is an interesting study but with some gaps. Some important clinical variables are missing.

  • All the controlors had a number of DNA copies under 300.000 and without suspected or diagnosed tissue-invasive disease?

 R: Yes, all the Controllers had a number of DNA copies under 300.000, and no one developed tissue-invasive disease (TID). (lines 76-77)

  • It’s important to know the donor and receptor CMV serological status.

R: Thank you for the suggestion. We have specified the CMV serological status (line 69-70).

  • Immunosuppressive regimen

Table 1: FK506 25 patients and tacrolimus 10 patients is the same drug ?!

Normally the KT patients are under triple immunosupression: CNI & MMF/MPA & steroid ; CNI & steroid & mTOR inhibitor for example – i don’t understand this table

R: Thank you for the suggestion. We have modified the table 1.

  • Other important variables

-Patients with end-organ CMV disease

-Patients with CMV syndrome

-Time to onset of CMV event

-Graft rejection or other event with change in Immunosuppressive regimen

R: Three patients had tissue-invasive disease (TID) and no patients had CMV syndrome (line 85)

HCMV infection was detected after median time of 48 days (IQR 35-73 days) in Controllers and 36 days (IQR 22-43 days) in Non-Controllers (lines 188-189).

Only one patient had Graft rejection (Line 86).

Reviewer 2 Report

Comments and Suggestions for Authors

Interesting work from Zavaglio et al. who found that during HCMV viremia, HCMV-specific T-cell immunity predicted self control of viremia without intervention and that weak immunity determined by CD4 CFC-pp65 assay was the best predictor of lack of self control of the infection.

Major comments:

1. I think the authors should define the study period during which these patients were identified.

2. I think it would be better to describe how many total SOT done during the study period, how many of these SOT were HCMV R+, and to describe how many of those did not develop HCMV reactivation/viremia.

3. Related to comment #2: Are these 40 patients include all the SOT HCMV R+ patients with viremia, or were there other with viremia at a very low level that the authors' protocol did not consider significant?

4. Page 2/11, line 67, how often patients are evaluated after the first 3 months post-transplant.

5. I think a major aspect is that we at the USA no longer use whole blood and copies/mL to quantify CMV viral load. For many years now we use plasma and IU/mL. I suggest the authors find a formula to convert their CMV viral load to what is equivalent in plasma by IU/mL to make it more relevant to current clinical practice. If authors are not able to do so, please explain.

6. In the manuscript, authors mentioned that HCMV-specific T-cell immunity assays were done at the peak HCMV-DNA. in other areas in the manuscript they mentioned that it is usually checked 2-3 months after transplant, please clarify. Also, please clarify how can treating teams or the local protocol predict that a specific HCMV-DNA at certain point is going to be the peak value and not the following value for example.

7. Were all 5 different assays performed on all 40 patients? and at the same date?

8. The discussion section expands on the description of ELISPOT and QauntiFERON and the differences between these two tests, however, I am not sure that this is the pertinent to the findings of the study. The paragraph in page 7, lines 222-249 can be curtailed.

9. The authors' protocol start IV GCV or PO VGCV when the viral is at a very high level of 300,000 copies/mL. Can the authors comment on whether their findings may suggest that viral load at a much lower level than 300,000 is associated with low HCMV-specific T-cell immunity, thus more likely to continue to replicate. 

Minor comments:

1. Figure 2. can be improved. For example I think CFC-iDC figures should A) B), CFC-iCL should be B) C), CFC-pp65 should be D) F), ELISPOT -iCL G), and ELISPOT -pp65 H). Right now the subfigures skip between different CFC assays and ELISPOT. In addition, I would add the p value to the corner of each subfigure. Right now it is hard to follow the p value for each assay. The asterisks mentioned in the footnote are very hard to see in the actual figures. Maybe adding the actual p value to each subfigure can avoid this ambiguity.

2. Same comment to table 2. CFC and ELISPOT assays are randomly mixed together in the table.

Thank you

Author Response

Comments and Suggestions for Authors

Interesting work from Zavaglio et al. who found that during HCMV viremia, HCMV-specific T-cell immunity predicted self control of viremia without intervention and that weak immunity determined by CD4 CFC-pp65 assay was the best predictor of lack of self control of the infection.

Major comments:

  1. I think the authors should define the study period during which these patients were identified.

R: The patients were enrolled Between June 2018 and December 2019. (Line 62)

  1. I think it would be better to describe how many total SOT done during the study period, how many of these SOT were HCMV R+, and to describe how many of those did not develop HCMV reactivation/viremia.

R: During the study 86 SOTR were enrolled. 75 of them were HCMV seropositive and 11 HCMV seronegative. Only 4 patients did not develop HMCV DNAemia (LINES from 62 to 65).

  1. Related to comment #2: Are these 40 patients include all the SOT HCMV R+ patients with viremia, or were there other with viremia at a very low level that the authors' protocol did not consider significant?

R: We have considered any viremia level. We enrolled in this study the patients for whom cells for T-cell response analysis were available

  1. Page 2/11, line 67, how often patients are evaluated after the first 3 months post-transplant.

R: According to local hospital protocols, HCMV DNAemia was quantified twice weekly by real-time PCR during the first three months after transplant. Later, patients were monitored for HCMV DNAemia at scheduled medical visits (at list once/month) or in the presence of HCMV-related clinical symptoms (Line from 70 to 74).

  1. I think a major aspect is that we at the USA no longer use whole blood and copies/mL to quantify CMV viral load. For many years now we use plasma and IU/mL. I suggest the authors find a formula to convert their CMV viral load to what is equivalent in plasma by IU/mL to make it more relevant to current clinical practice. If authors are not able to do so, please explain.

R: We added the IU/mL in the methods and results paragraph (Lines 84, 192-193).

  1. In the manuscript, authors mentioned that HCMV-specific T-cell immunity assays were done at the peak HCMV-DNA. in other areas in the manuscript they mentioned that it is usually checked 2-3 months after transplant, please clarify. Also, please clarify how can treating teams or the local protocol predict that a specific HCMV-DNA at certain point is going to be the peak value and not the following value for example.

R: We clarified this information in lines 86 and 202

  1. Were all 5 different assays performed on all 40 patients? and at the same date?

R: Yes, the 5 different assays were performed on all 40 patients and at the same date.

  1. The discussion section expands on the description of ELISPOT and QauntiFERON and the differences between these two tests, however, I am not sure that this is the pertinent to the findings of the study. The paragraph in page 7, lines 222-249 can be curtailed.

R: ok, we curtailed the text.

  1. The authors' protocol start IV GCV or PO VGCV when the viral is at a very high level of 300,000 copies/mL. Can the authors comment on whether their findings may suggest that viral load at a much lower level than 300,000 is associated with low HCMV-specific T-cell immunity, thus more likely to continue to replicate. 

R: Patients with low HCMV DNA level may have a low HCMV-specific T-cell response, thus it is likely that virus continue to replicate, or a high T-cell response able to control further virus replication. We added this comment in the end of the discussion (line 370).

Minor comments:

  1. Figure 2. can be improved. For example I think CFC-iDC figures should A) B), CFC-iCL should be B) C), CFC-pp65 should be D) F), ELISPOT -iCL G), and ELISPOT -pp65 H). Right now the subfigures skip between different CFC assays and ELISPOT. In addition, I would add the p value to the corner of each subfigure. Right now it is hard to follow the p value for each assay. The asterisks mentioned in the footnote are very hard to see in the actual figures. Maybe adding the actual p value to each subfigure can avoid this ambiguity.

R: Thank you for this suggestion. We corrected the figure 6 as suggested

  1. Same comment to table 2. CFC and ELISPOT assays are randomly mixed together in the table.

R: Thank you for this suggestion. We corrected the table 3 as suggested

Thank you

Reviewer 3 Report

Comments and Suggestions for Authors

The authors here have followed up on their previous publication and precisely studied role of HCMV-specific T-cell immunity measured at time of HCMV-DNA peak to predict the spontaneous clearance of HCMV infection.

Overall, Manuscript is well written with logical experiments. 

I have few minor comments:
-It would be ideal to show a representative dot plot for intra cellular cytokine assay and a representative image of Elispot assay. 

-It would be also recommended to include a gating strategy for the flow cytometry assays. 

Few grammatical corrections:

Line 130 - There seem to be an extra space before the sentence "Spots were....".

 Line 175 - Starts with "In details". However, it should be "In detail:. 

Author Response

Comments and Suggestions for Authors

The authors here have followed up on their previous publication and precisely studied role of HCMV-specific T-cell immunity measured at time of HCMV-DNA peak to predict the spontaneous clearance of HCMV infection.

Overall, Manuscript is well written with logical experiments.

R:  Thank you very much for your comment.

I have few minor comments:

1) It would be ideal to show a representative dot plot for intra cellular cytokine assay and a representative image of Elispot assay.

2) It would be also recommended to include a gating strategy for the flow cytometry assays.

R: We added the gating strategy and representative dot plot for the flow cytometry assays (Figure 1) and the representative image of ELISPOT assay (Figure 2)

Few grammatical corrections:

Line 130 - There seem to be an extra space before the sentence "Spots were....".

R: We have corrected

Line 175 - Starts with "In details". However, it should be "In detail:.

R: We have corrected

Reviewer 4 Report

Comments and Suggestions for Authors

The manuscript of Zavaglio et al compares different immunological procedures to estimate the size of HCMV-specific T lymphocytes, with the scope of predicting the capability of solid organ transplant recipients to control HCMV infection. The results evidentiate the better predictive value of the size of antigen-specific CD4+ rather than CD8+ T cells, and show the informativeness of flow cytometry-based techniques, that have the advantages of providing a response in short time and to be more readily available in clinical settings. Regarding the procedure used to obtain the flow cytometric information, the authors have transformed the percentage of IFN-g+ antigen-specific CD4+ (or CD8+) T cells in the absolute number of IFNg-producing cells. What are the reasons that have led to choose this strategy? Please provide the methodological information on how the absolute number of CD4+ and CD8+ T cells were obtained. Would the comparison of the percentage of IFNg+ CD4+ and CD8+ between controllers and non controllers provide the same information? If the data need to be presented in the transformed version, the authors should show the absolute numbers of total CD4+ and CD8+ T cells in controller and non controller groups.

Minor points:

Materials and methods:

1. please add information on the timelength of the stimulation (with iDC, pp65pool, ICL) for flow cytometric assay

2. lines 95-99 report a duplicated sentence, with two different cited references; please correct.

Figures 2: please enlarge the character size to improve legibility

Comments on the Quality of English Language

Minor corrections in the english and spelling would improve the text

Author Response

Comments and Suggestions for Authors

The manuscript of Zavaglio et al compares different immunological procedures to estimate the size of HCMV-specific T lymphocytes, with the scope of predicting the capability of solid organ transplant recipients to control HCMV infection. The results evidentiate the better predictive value of the size of antigen-specific CD4+ rather than CD8+ T cells, and show the informativeness of flow cytometry-based techniques, that have the advantages of providing a response in short time and to be more readily available in clinical settings.

  • Regarding the procedure used to obtain the flow cytometric information, the authors have transformed the percentage of IFN-g+ antigen-specific CD4+ (or CD8+) T cells in the absolute number of IFNg-producing cells. What are the reasons that have led to choose this strategy?

R: we calculated the absolute number to take into account both the percentage of HCMV-specific T cells and the number of circulating lymphocytes.

  • Please provide the methodological information on how the absolute number of CD4+ and CD8+ T cells were obtained.

R: The absolute number CD3+CD4+ and CD3+CD8+ T-cell count was measured in whole blood using flow cytometry (BD MultitestTM CD3/CD8/CD45/CD4 with BD TruCOUNTTM Tubes, BD Biosciences). The total number of HCMV-specific CD4+ and CD8+ T-cells was then calculated by multiplying the percentage of IFN-γ producing HCMV-specific T-cells by the corresponding absolute CD4+ and CD8+ T-cell count (line from 124 to 129).

  • Would the comparison of the percentage of IFNg+ CD4+ and CD8+ between controllers and non controllers provide the same information?

R: Yes, the percentage of IFNg+ CD4+ and CD8+ between Controllers and Non-Controllers provide the same information. We specified this in the results paragraph

  • If the data need to be presented in the transformed version, the authors should show the absolute numbers of total CD4+ and CD8+ T cells in controller and non controller groups.

R: We added the figure regarding the absolute number of total CD4+ and CD8+ T cells in Controllers and Non-Controllers (Figure 3)

Minor points:

Materials and methods:

  1. please add information on the timelength of the stimulation (with iDC, pp65pool, ICL) for flow cytometric assay

R: We added information on the timelenght of the stimulation (with iDC, pp65pool, ICL) for flow cytometry assay at line 104.

  1. lines 95-99 report a duplicated sentence, with two different cited references; please correct.

R: We have corrected the citation at Lines 106-108

  1. Figures 2: please enlarge the character size to improve legibility

R: We have enlarged the character size of figure 2

Comments on the Quality of English Language

Minor corrections in the english and spelling would improve the text

Round 2

Reviewer 1 Report

Comments and Suggestions for Authors

Thanks for the changes.

Author Response

Thank you!

Reviewer 2 Report

Comments and Suggestions for Authors

I thank the authors for their detailed response.

Minor 

1. Page 2, line 73: at least instead of at list. 

Major:

1. Please add the reasoning for enrolling 40/75 HCMV seropositive recipients because these 40 had available CMV-specific T cell immunity testing available. 

Thank you

Comments on the Quality of English Language

The Englis is very good, I only detected one error mentioned in the minor comments. 

Author Response

Comments and Suggestions for Authors

I thank the authors for their detailed response.

Minor

  1. Page 2, line 73: at least instead of at list.

R: We have corrected

Major:

  1. Please add the reasoning for enrolling 40/75 HCMV seropositive recipients because these 40 had available CMV-specific T cell immunity testing available.

R: We have added this information (Line 67).

Thank you

Comments on the Quality of English Language

The English is very good, I only detected one error mentioned in the minor comments.

Reviewer 4 Report

Comments and Suggestions for Authors

Materials and methods:

Please clarify these seemingly conflicting sentences:

Lines 65-67: 40 HCMV seropositive SOTR, including 27 kidney transplant recipients (KTR), 6 lung transplant recipients (LTR) and 7 heart transplant recipients (HTR) were enrolled in this study

Lines 69-79: HCMV serological status was positive in 22 donors, negative in 13 donors and unknown in 4 donors.

Comments on the Quality of English Language

minor editing of the text would be advisable

Author Response

Comments and Suggestions for Authors

Materials and methods:

Please clarify these seemingly conflicting sentences:

Lines 65-67: 40 HCMV seropositive SOTR, including 27 kidney transplant recipients (KTR), 6 lung transplant recipients (LTR) and 7 heart transplant recipients (HTR) were enrolled in this study

Lines 69-79: HCMV serological status was positive in 22 donors, negative in 13 donors and unknown in 4 donors.

R: We have clarified these sentences.

Comments on the Quality of English Language

minor editing of the text would be advisable